# Porcine Circovirus Type 2 Induces Single Immunoglobulin Interleukin-1 Related Receptor (SIGIRR) Downregulation to Promote Interleukin-1β Upregulation in Porcine Alveolar Macrophage

**DOI:** 10.3390/v11111021

**Published:** 2019-11-03

**Authors:** Shunli Yang, Baohong Liu, Shuanghui Yin, Youjun Shang, Xinming Zhang, Muhammad Umar Zafar Khan, Xiangtao Liu, Jianping Cai

**Affiliations:** 1State Key Laboratory of Veterinary Etiological Biology, National Foot and Mouth Disease Reference Laboratory, Innovative Team for GI Infection and Mucosal Immunity of Swine and Poultry, Lanzhou Veterinary Research Institute, Chinese Academy of Agricultural Sciences, Lanzhou 730046, China; yangshunli@caas.cn (S.Y.); liubaohong@caas.cn (B.L.); shangyoujun@caas.cn (Y.S.); zxm10202114@163.com (X.Z.); umarzafar8@caas.cn (M.U.Z.K.); liuxiangtao@caas.cn (X.L.); 2Jiangsu Co-Innovation Center for Prevention and Control of Important Animal Infectious Diseases and Zoonoses, Yanzhou 225009, China

**Keywords:** porcine circovirus type 2, porcine alveolar macrophage, inflammatory cytokines, single-immunoglobulin interleukin-1 related receptor (SIGIRR), IL-1β

## Abstract

Multisystemic inflammation in pigs affected by porcine circovirus type 2 (PCV2) indicates the disordered expression of inflammatory cytokines. However, the PCV2-induced expression profile of inflammation cytokines and its regulating mechanism remain poorly understood. In this study, inflammatory cytokines and receptors in porcine alveolar macrophages (PAMs) after PCV2 infection were profiled in vitro by an RT^2^ Profiler^TM^ PCR array assay. The regulatory mechanism of interleukin-1β (IL-1β) expression was investigated. Results showed that 49 of 84 inflammation cytokines and receptors were differentially expressed (*p* < 0.05, absolute fold change ≥2) in PAMs at different stages post-PCV2 infection. Moreover, the overexpression of single-immunoglobulin interleukin-1 related receptor (SIGIRR) or the blocking of NF-κB activation by its inhibitor markedly decreased IL-1β secretion. This finding suggested that PCV2-induced overexpression of IL-1β was associated with the downregulation of SIGIRR and the activation of NF-κB. Furthermore, the excessive activity of NF-κB in SIGIRR-knockout PAMs cell line, indicating that SIGIRR negatively regulated IL-1β production by inhibiting the activation of NF-κB. Overall, PCV2-induced downregulation of SIGIRR induction of NF-κB activation is a critical process in enhancing IL-1β production in PAMs. This study may provide insights into the underlying inflammatory response that occurs in pigs following PCV2 infection.

## 1. Introduction

Porcine circovirus 2 (PCV2) belongs to the genus *Circovirus* of the family Circoviridae and is known as the primary cause of PCV2-systemic disease (PCV2-SD), which is initially described as a postweaning multisystemic wasting syndrome (PMWS) [1,2]. PCV2-infected piglets easily develop secondary infections, such as porcine reproductive and respiratory syndrome virus (PRRSV), porcine parvovirus (PPV), classical swine fever virus (CSFV), pseudorabies virus (PRV), and Mycoplasma hyopneumoniae [3,4], suggesting that PCV2 associated diseases (PCVAD) are actually immunosuppressive diseases.

Granulomatous inflammation of lymph nodes is one of the most remarkable gross findings observed in PCV2-SD pigs. The histopathology of the lymphoid tissues in those pigs showed severe lymphoid depletion accompanied by diffuse infiltration of histiocytic cells, indicating the potential involvement of an inflammatory response [5]. Other inflammatory lesions, such as interstitial pneumonia, interstitial nephritis, granulomatous enteritis, and periportal hepatitis were observed in pigs with PCVAD [6]. PCV2 can induce strong IL-1β and IL-8 responses in peripheral blood mononuclear cells (PBMCs) [7], which is a fact consistent with the chronic inflammatory nature of PCV2-SD. PCV2 is known to target immune cell subsets [8,9], including porcine alveolar macrophages (PAMs) [10], which are the first line of the pulmonary defense system against various pathogens [11]. The lungs of pigs suffering from PCV2-associated respiratory disease show elevated levels of IL-1β and IL-8 mRNA, the same finding could be expected in a pig suffering from interstitial pneumonia [12]. However, the expression profile of inflammatory cytokines in PAMs after PCV2 infection and the mechanism through which PCV2 induces PAMs to produce inflammatory cytokines are largely unknown.

Interleukin-1R like receptors (IL-1Rs) and Toll-like receptors (TLRs) are key receptors of innate immunity and inflammation. These receptors are members of a superfamily of phylogenetically conserved proteins characterized by the presence of a conserved intracellular domain, the Toll/IL-1R domain [13]. Single-immunoglobulin interleukin-1 related receptor (SIGIRR), which is also known as Toll/IL-1R8 (TIR-8), is a member of the ILRs family [13,14]. SIGIRR is a regulatory protein acting intracellularly to inhibit ILR and TLR signaling [15,16]. SIGIRR-deficient mice exhibited a severe intestinal inflammation compared with wild type mice in response to dextran sodium sulfate, which was associated with increased leukocyte infiltration and proinflammatory cytokines (TNF-a, IL-6, IL-1β, IL-12p40, IL-17, MIP-2/CXCL1, KC/CXC, and JE/CCL2) [17,18,19]. The association of chronic inflammation induced by PCV2 with the expression of SIGIRR remains unclear. In this study, the expression profile of inflammatory cytokines in PAMs inoculated with PCV2 was analyzed using an RT^2^ Profiler^TM^ PCR array assay. The expression of SIGIRR and its function in the process of PCV2-induced IL-1β expression were further investigated. Our results showed that a large number of inflammatory cytokines were differentially expressed in PCV2-infected PAMs in vitro. Furthermore, the up-regulation of IL-1β in PCV2-infected PAMs was related to the excessive activity of NF-κB which resulted from the down-regulation of SIGIRR. 

## 2. Materials and Methods

### 2.1. Cell Culture and Virus Preparation

PAMs were obtained by lung lavage [20] from three 5-week-old specific-pathogen-free landrace pigs which tested negative for PCV1, PCV2, swine fever virus, porcine reproductive and respiratory syndrome virus, pseudorabies virus, and porcine parvovirus infections. In brief, alveolar lavage of the lungs was performed with 200 mL aliquots of sterile phosphate buffer saline containing 100 U/mL penicillin and 100 µg/mL streptomycin. PAMs were obtained by centrifuging lavage fluids at 500× *g* for 10 min, and the cells were washed with RPMI-1640 medium (Thermo Scientific, USA) three times. To confirm the numbers of living PAMs, the PAMs were stained with 7-AAD (BD Pharmingen, USA) and analyzed by flow cytometry. The concentration of PAMs was adjusted to 1 × 10^7^ cells/mL (Living cells) with a growth medium containing RPMI-1640 supplemented with 10% (*v*/*v*) fetal bovine serum (FBS) (Gibco, Australia). PAMs were seeded in a 25 cm^2^ cell culture flask (Corning Costar, USA) and incubated at 37 °C under 5% CO_2_.

PCV-free PK-15 cells (PK-15 ATCC® CCL-33™ USA) were maintained in DMEM supplemented with 10% heat-inactivated FBS, 100 U/mL penicillin, and 100 µg/mL streptomycin. PAMs cell line 3D4/2 (CRL-2845) [21] from American Type Culture Collection (USA) has been used as a research model of PCV2 infection [22]. 3D4/2 were cultured in RPMI-1640 medium supplemented with 10% FBS, 100U/mL penicillin, and 100 µg/mL streptomycin.

A PCV2b strain, ShDNY-PCV2 (GenBank No. FJ948167), originally isolated from the lung of a pig with respiratory disease, was propagated in PK-15 cell monolayer according to the methods reported previously [23]. The titer of the PCV2 inoculum was 1.0 × 10^5^ TCID_50_ per mL as determined by titration in PCV-free PK-15 cells.

### 2.2. Infection, RNA Extraction, and cDNA Synthesis

PAMs generated from pigs were inoculated with PCV2b at a multiplicity of infection (MOI) of 1 and incubated at 37 °C under 5% CO_2_, for 1 h. The supernatant was removed, and the cells were washed with RPMI-1640 medium three times. The cells were then cultured in a maintain medium containing RPMI-1640 supplemented with 2% FBS and incubated at 37 °C under 5% CO_2_ for 1, 24, and 48 h. The lysate of PCV2-free PK-15 cells was used as a control to counteract gene expression changes induced by PK-15 cells rather than being a true result of PCV2 infection. The PAMs and supernatant were collected at indicated time intervals. Cell samples were collected by centrifugation of 500× *g* for 10 min, then cell pellet was kept in RNAlater^®^ RNA stabilization reagent kits (Qiagen, Germany) to prevent RNA degradation and stored at −70 °C until use. The supernatant was centrifuged at 500× *g* for 10 min and stored at −70 °C for further analysis.

Total RNA was extracted using an RNeasy plus mini kit (Qiagen, Germany) according to the manufacturer’s instruction. The quantity of RNA was evaluated with a Qubit 4 fluorometer (Invitrogen by Thermo Fisher Scientific, USA) using the Qubit RNA HS assay kit. The RNA quality and integrity were then evaluated using the Qubit RNA IQ assay kit. 5 µg of RNA of each sample was reverse-transcribed to cDNA by using an RT^2^ HT first strand kit (Qiagen, Germany) and suspended in a final solution of 110 µL. All cDNA samples were stored at −20 °C for further real-time quantitative PCR assay.

### 2.3. The Real-Time Quantitative PCR Assay

The number of PCV2 copies per milliliter of PCV2-inoculated PAMs was determined by comparison with known standards by using quantitative PCR (qPCR) as previously described [24]. DNA was extracted from 200 µL of the PCV2-inoculated PAMs using a DNA extraction kit (DNeasy Tissue Kit; Qiagen, Germany) and eluted in 30 µL of buffer according to the manufacturer’s instruction. 3 µL of DNA was detected in a total of 25 µL SYBR green qPCR reaction buffer. A plasmid containing the PCV2 genome was used as the standard.

An RT^2^ Profiler PCR array (PASS-011ZA, Qiagen, Germany) was used to analyze pig inflammatory cytokines and receptors according to the manufacturer’s protocol. Real-time qPCR was performed in an Mx3005p cycler (Agilent Technologies, USA). The mRNA expression levels obtained for each gene of interest (GOI) were normalized by the housekeeping gene through the following equation: ΔCt = Ct(GOI) − Ct (housekeeping gene), where Ct is the threshold cycle. Each reported GOI represented the mean increase or decrease in mRNA expression relative to the levels of the controls. Actin-beta (Actin-β), Beta-2-microglobulin (B2M), Glyceraldehyde-3-phosphate dehydrogenase (GAPDH), Hypoxanthine phosphoribosyltransferase (HPRT1), and Ribosomal protein L13a (RPL13A) were used as housekeeping genes. Relative expression intensity was obtained by calculating 2^−ΔΔCt^ for each sample from PCV2- and mock-infected control PAMs. 

The expression of SIGIRR in PCV2-infected PAMs at the mRNA level was detected by real-time qPCR using primer SIGs: 5′-TCGAGGGCCAGCGACGCGA-3′, and SIGa: 5′-AGCGCCA GCTGTAGCTCTT-3′. IL-1β was detected using primer IL-1Bs: 5′-GTGTTCTGCATGAGCTTTGTGC-3′, and IL-1Ba: 5′-ACCAGTTGGGGTA CAGGGCAGA-3′. Actin-β was used as the reference gene [25]. PCR was performed under the following cycling conditions: initial denaturation at 95 °C for 3 min, followed by 40 cycles of denaturation at 95 °C for 10 s, annealing at 58 °C for 30 s, extension at 72 °C for 20 s. mRNA were normalized by comparing to actin-β and expressed relative to mock control. Triplicate samples were tested by real-time qPCR.

### 2.4. Analysis of Cytokine Expression in Protein Level

The protein expression levels of IL-1β, IL-10, IL-8, granulocyte-macrophage colony-stimulating factor (GM-CSF), and tumor necrosis factor alfa (TNF-α), which were significantly changed at the mRNA level as shown by the PCR array assay, were determined by testing the culture supernatants with a RayBiotech fluorescent Quantibody^®^ porcine cytokine array system (RayBiotech, China). Array slides were treated and processed according to the manufacturer’s instructions. Triplicate samples were tested by cytokine array system.

### 2.5. Western Blot Analysis

PCV2 in PAMs after the infection was detected through Western blot assay using PCV2 antibody-positive porcine serum that was collected from virus-like particles (VLPs) immunized pigs [26]. The expression of SIGIRR and NF-κB in PCV2-infected PAMs was evaluated by Western blot based on the anti-SIGIRR polyclonal antibody (Abcam, USA), anti-NF-κB p-p65 (phosphorylation S536) polyclonal antibody (Abcam, USA), and anti-NF-κB p65 monoclonal antibody (Abcam, USA). PAMs were infected with PCV2 at an MOI of 1 for 24 h. Total cell extracts were prepared in NP-40 lysis buffer supplemented with protease and phosphatase inhibitors. The cell lysates were separated by 12% SDS-PAGE and transferred onto 0.22 µm PVDF membranes (Millipore, USA). The membranes were blocked with 5% (*w*/*v*) BSA in Tris-HCl buffer saline with Tween-20 (TBST) containing 100 mM NaCl, 25 mM Tris (pH 7.6), and 0.1% Tween-20. The membranes were incubated with primary antibodies against Cap, SIGIRR, p-p65, p65, and β-actin (Proteintech, USA). After washing with TBST for five times, the membranes were incubated with appropriate (anti-pig/-mouse/-rabbit) horseradish peroxidase-conjugated secondary antibodies (Abcam, USA). Proteins were visualized using the ECL reagent (Thermofisher, USA) according to the manufacturer’s instructions.

### 2.6. Inhibition of Signal Transduction Pathway

BAY11-7082 (Selleckchem, USA), an inhibitor of NF-κB, by inhibiting the NF-κB p65 nuclear translocation and DNA binding activity, has been used in the research of NF-κB signaling pathway [25,27,28]. In this study, PAMs were pretreated with DMSO, BAY11-7082 (1 µM) for 1 h, as described previously [28], and infected with PCV2 at an MOI of 1. The cells and the supernatants were harvested after 12 h for analysis of IL-1β mRNA and protein by real-time RT-PCR and ELISA, respectively.

### 2.7. SIGIRR Overexpression and Knocked off in 3D4/2 Cells

Three guide RNAs (g1, g2, and g3) were designed based on the NCBI reference sequence (NC_010444.3) and porcine SIGIRR sequence cloned from PAMs (GenBank No. MN355505). The guide RNAs were cloned into a lentiCRISPRv2 plasmid (Miaolingbio, China). The gene of SIGGIR (GenBank No. MN355505) was coned into the pCDH-CMV-MCS-PURO plasmid (Miaolingbio, China). The recombinant plasmids were transfected to PAMs cell line 3D4/2 cells. After 48 h of culture, 2 µg/mL puromycin was added to the cell cultures to select the recombinant plasmid-positive cells. The selected cells were checked by genomic PCR sequencing, immunofluorescence assay (IFA), and Western blot assay.

### 2.8. Immunofluorescence Assay (IFA)

PCV2 in PAMs after the infection was detected by IFA based on quantum dots-conjugated single-domain antibody probe against the Cap protein. Stained cells were visualized using an inverted fluorescence microscope (Leica, Germany) at a wavelength of 605 nm (red) [29,30]. The nucleus was stained with DAPI (blue). The expression of SIGIRR in its upregulated and knocked off 3D4/2 cells was checked by IFA based on anti-SIGIRR antibody and FITC labeled anti-rabbit antibody (Abcam, USA). 

### 2.9. ELISA

The levels of secreted IL-1β in the cell culture supernatants were measured using a commercially available porcine IL-1β ELISA kit (R&D Systems, USA) according to the manufacturer’s instruction.

### 2.10. Statistic Analysis

All data are expressed as mean ± SD. PCR array data were analyzed using RT^2^ Profiler PCR array data analysis software version 3.5 (Qiagen, Germany). Statistical differences in data were evaluated by Student’s *t*-test. A *p*-value of less than 0.05 was considered significant.

### 2.11. Ethics Statement

All of the animals were handled in strict accordance with the Animal Ethics Procedures and Guidelines of the People’s Republic of China. This study was approved by the Animal Ethics Committee of Lanzhou Veterinary Research Institute, Chinese Academy of Agricultural Sciences (No. LVRIAEC2012-006, 1 January 2017).

## 3. Results

### 3.1. Confirmation of PCV2b Infection in PAMs

The IFA (Figure 1A) and Western blot (Figure 1B) results revealed red fluorescent cells and Cap protein bands, respectively, in the infected PAMs, indicating that the PAMs were successfully infected by PCV2 at 24 and 48 h post-infection (hpi). Meanwhile, qPCR revealed the presence of nucleic acid of PCV2b in PAMs at 1, 24, and 48 hpi, but the levels of viral DNA were not significantly different among the three-time points (Figure 1C).

### 3.2. mRNA Expression Profiles

PCV2b-infected PAMs were analyzed by using a PCR array kit to investigate the expression profile of inflammatory cytokines in the interaction of PCV2 with host cells. Actin-β that doesn’t show the significant change between different groups and time points was used as a normalized reference gene. Forty-nine out of eighty-four inflammatory cytokines and receptors were differentially expressed (*p* < 0.05, absolute fold change ≥ 2) in different stages of PCV2b-infected PAMs (Appendix A). In these genes, several interleukins, chemokines, chemokine receptors, tumor necrosis factors, and colony-stimulating factors were differentially expressed at different time points after PCV2b infection (Figure 2).

### 3.3. Expression Analyses of Several Cytokines in PCV2b-Infected PAMs

The PCR array analysis showed that IL-1β, IL-8, IL-10, GM-CSF, and TNF-α were significantly upregulated in PAMs infected with PCV2b. These cytokines were subjected to a Quantibody^®^ porcine cytokine array to examine if gene expression led to protein expression changes. The results demonstrated that the changes in the expression trend of these cytokines are consistent with the results of the PCR array analysis (Figure 3).

### 3.4. Downregulation of SIGIRR Expression in PCV2b-Infected PAMs

The expression of SIGIRR, a negative regulator of toll-like receptors mediated immune responses, was analyzed in PCV2-infected PAMs. The expression of SIGIRR was downregulated at 1 and 12 hpi, as indicated by the qPCR and Western blot results (Figure 4).

### 3.5. NF-κB Activation Is Involved in PCV2b-Induced IL-1β Secretion

PCV2 can activate NF-κB signaling [25,28,31] and PCV2-induced IL-1β expression in PAMs is related to the activation of the NF-κB signaling pathway [32]. The phosphorylation of p65 during PCV2 infection was analyzed by Western blot based on phospho-specific antibody, p-p65. Results showed that the phosphorylation of p65 was detected at different time points after PCV2 infection (Figure 5A). Cells were pretreated with an inhibitor BAY11-7082 then infected with PCV2 to determine whether activation of NF-κB is essential for PCV2-induced IL-1β production. BAY11-7082 significantly suppressed the production of IL-1β in PCV2-infected PAMs, indicating the blocked NF-κB activation inhibited PCV2-induced IL-1β expression (Figure 5B,C).

### 3.6. SIGIRR Is Associated with PCV2b-Induced IL-1β Secretion

The expression of IL-1β, SIGIRR, and the activation of NF-κB in a PAMs cell line 3D4/2 post PCV2 infection were investigated by qPCR and Western blot assay. The results showed that the expression trend of IL-1β and SIGIRR, and the activation of NF-κB were similar to it in primary PAMs (Appendix A). Thus, the 3D4/2 cell was used to investigate the mechanism of PCV2-induced upregulation of IL-1β.

The SIGIRR was knocked out in 3D4/2 by the CRISPR-Cas9 system. Three gRNAs were designed to target three different sites of the SIGIRR gene (Figure 6A), and three cell clones (g1, g2, and g3) were selected and identified. The sequencing results showed the targeted sequences had deleted four and six nucleic acid-base pairs in cell g1 and g3, respectively (Figure 6B). SIGIRR was not detected in g1 cells but expressed in g2 and g3 cells by Western blot assay (Figure 6C). In g3 cells, the deleted six-base pairs do not have an effect on the protein translation because it does not disrupt the codon sequence but only removes the amino acid Ser and Asp of SIGIRR.

The SIGIRR was overexpressed by a pCDH-SIGIRR plasmid to determine whether the downregulation of SIGIRR is associated with the significant upregulation of IL-1β. SIGIRR was upregulated significantly in pCDH-SIGIRR plasmid-transfected cells compared with wild 3D4/2 cells by Western blot (Figure 6D). In addition, IFA showed that SIGIRR was a higher expression in pCDH-SIGIRR plasmid transfected cells but was not expressed in g1 cells (Figure 6E).

PCV2b was inoculated with SIGIRR-knockout (SIGIRR-/-), SIGIRR-overexpression (SIGIRR+/+), and wild-type cells at 1 MOI for 12 h. IL-1β was analyzed by qPCR at the mRNA level, and secreted IL-1β was detected by ELISA. Results showed that IL-1β was upregulated significantly in the SIGIRR-knockout cells and wild-type cells, but not in SIGIRR-overexpressed cells (Figure 6F,G).

### 3.7. SIGIRR Is Involved in PCV2-Induced NF-κB Activation

The SIGIRR+/+ and SIGIRR-/- cells were infected with PCV2, and the activation was detected by Western blot assay. The results showed increased phosphorylation of p65 in SIGIRR-/- cells compared with that in SIGIRR+/+ cells (Figure 7). These results indicated that the insufficiency of SIGIRR resulted in exaggerated NF-κB activation, which contributed to excessive IL-1β expression.

## 4. Discussion

PCV2, one of the most important pathogens in domestic swine worldwide, is the causative agent of PCV-SD. Multisystemic inflammatory lesions, including hepatitis, dermatitis, enteritis, pneumonia, and lymphadenitis [33,34], observed in PCV2-affected pigs, revealed that PCV2 can induce the excessive expression of inflammatory cytokines. In this study, 84 of porcine inflammatory cytokines and receptors, including interleukins, colony-stimulating factors, chemokines, chemokine receptors, and tumor necrosis factors, which are responsible for inflammation were analyzed by the RT² Profiler™ PCR Array system. Forty-nine of 84 of these genes were differentially expressed in PAMs at the early stage after PCV2 infection. Hence, the inflammatory cytokines were possibly implicated in the pneumonia of PCV2 affected pigs.

PCV2 has been shown to infect and persist in cells of the immune system, including PAMs [10,35]. The presence of PCV2 in alveolar macrophages seems to induce differential expression of some cytokines [5,35]. Similar to described in previous reports [32,36], in this study PCV2b, a pathogenic strain currently circulating in China, was present in PAMs in the absence of clear viral replication that was indicated by IFA, Western blotting, and qPCR assay. Thus, the identification of the expression profile of cytokines in the interaction process of PCV2 with PAMs may provide insights into the molecular mechanisms of inflammation.

CSF has fundamental roles in driving and regulating inflammatory responses by regulating the proliferation and differentiation of neutrophils, monocytes, and macrophages [37,38,39]. In addition, GM-CSF is required for alveolar macrophages to maintain surfactant protein homeostasis in the alveolar spaces [40] and for macrophage-based tissue repair [41]. In this study, GM-CSF was upregulated in PCV2-infected PAMs, suggesting that GM-CSF could possibly regulate inflammation, maintain PAMs function, and promote tissue repair in the interaction of PCV2 with the host.

Multisystemic granulomatous inflammation accompanied by multinucleated giant cells, histiocytes, and macrophage infiltration is a common histopathological finding in pigs with PCV-SD [33]. Recruitment of inflammatory cells into injured tissue is regulated by various cytokines and chemokines. MCP-1 is known to be an important chemokine for the recruitment of monocytes from the blood [42], which was suggested to play an important role in the pathogenesis of granulomatous inflammation in pigs with PCV-SD [43,44,45]. The chemotaxis of MCP-1 can be further enhanced by IL-8 [46] which is a chemoattractant for neutrophils and other polymorphonuclear leukocytes that are produced following acute infection [47]. In this study, both MCP-1 and IL-8 were upregulated in PCV2-infected PAMs. In addition, chemokines such as CCL8, CCL17, and CXCL2 were upregulated in PAMs at the early stage after PCV2 infection. Thus, macrophage infiltration has been suggested as an important feature of the pathogenesis and progression of PMWS [48].

IL-1β and TNF-α are acute-phase proinflammatory cytokines that promote inflammation and induce fever, and tissue destruction [49]. Our study showed that IL-1β and TNF-α were upregulated significantly in PCV2-infected PAMs, which have similar expression trends in lymphoid tissues and PBMCs of pigs with PCV-SD [7,50]. In addition, cytokines, including IL-1α, IL-6, IL-18, IL-23, TNFSF10, and TNFSF13, were upregulated to a different extent in PCV2-infected PAMs. These cytokines increased expression was in concordance with the nature of systemic inflammation in PCV2 affected pigs.

The differentially expressed IL-1β, IL-10, IL-8, GM-CSF, and TNF-α were analyzed at the protein level with a Quantibody array assay to verify whether the differentially expressed gene leads to its protein expression changes. The expression of IL-1β, IL-10, IL-8, GM-CSF, and TNF-α fitted well at mRNA and protein levels. In addition, the expression trend of IL-1β, IL-10, IL-8, and TNF-α in PCV2-infected PAMs are in good agreement with the previous reports [22,27,28,32,36].

SIGIRR has been suggested to function as a blocking receptor because of its interaction with key signaling molecules (MyD88, IRAK, and TRAF6) of the IL-1R/TLR-NF-κB signaling pathway [16,51]. IL-1β was upregulated significantly in SIGIRR-deficient mice with necrotizing enterocolitis [52]. SIGIRR may be a key regulator of inflammation, which in physiological conditions inhibits activation of IL-1R/TLR to avoid detrimental inflammation, while during inflammatory responses or in pathological conditions SIGIRR is downregulated thereby permitting IL-1R/TLR activation [14,18,53,54]. In the present study, SIGIRR was down-regulated in PAMs at the early stage (1 and 12 hpi) after PCV2 infection, accompanied by the overexpression of inflammatory cytokines, especially IL-1β (significantly in mRNA and protein level). SIGIRR was overexpressed and knocked off in a 3D4/2 cell line to investigate its role in PCV2-induced overexpression of IL-1β. IL-1β was found significantly upregulated in SIGIRR-/- cells but was not in SIGIRR+/+ cells, suggesting SIGIRR could be involved in the PCV2-induced upregulation of IL-1β.

NF-κB is a key signaling molecule in the TLRs/NF-κB pathway. NF-κB phosphorylation and nuclear translocation after PCV2 infection have been reported [25,31,55]. Several cytokines, including IL-1β, in PCV2-infected PAMs, were upregulated through the activation of NF-κB [22,28,32]. In this study, NF-κB was activated in PCV2-infected PAMs, and the blocking of NF-κB by its inhibitor markedly decreased PCV2-induced IL-1β secretion. The results suggested that PCV2-induced upregulation of IL-1β occurred through the activation of NF-κB, that was in concordance with the previous report [32]. To determine the role of SIGIRR in NF-κB activation, we measured the phosphorylation of NF-κB in SIGIRR+/+ and SIGIRR-/- cells with PCV2 infection. SIGIRR-/- cells significantly increased activation of NF-κB compared with that of SIGIRR+/+ cells, suggesting that overexpression of IL-1β in PCV2-infected PAMs is due to an exaggerated activation of NF-κB caused by the inability of SIGIRR to inhibit NF-κB activation.

## 5. Conclusions

Forty-nine inflammatory cytokines and receptors were differentially expressed in PAMs post PCV2 infection. The results suggested that the inflammatory cytokines were possibly implicated in the pneumonia of the PCV2 affected pigs. The upregulation mechanism of IL-1β was investigated in the 3D4/2 cell line. The results revealed that the increased IL-1β expression was related to the excessive activation of NF-kB caused by the downregulation of SIGIRR. Our study indicated that SIGIRR may play a role in the increased expression of inflammatory cytokines in PCV2-infected PAMs. These findings contribute to understanding the mechanism underlying cytokines-mediated response associated with the chronic inflammatory nature of PCV-SD.

## Figures and Tables

**Figure 1 viruses-11-01021-f001:**
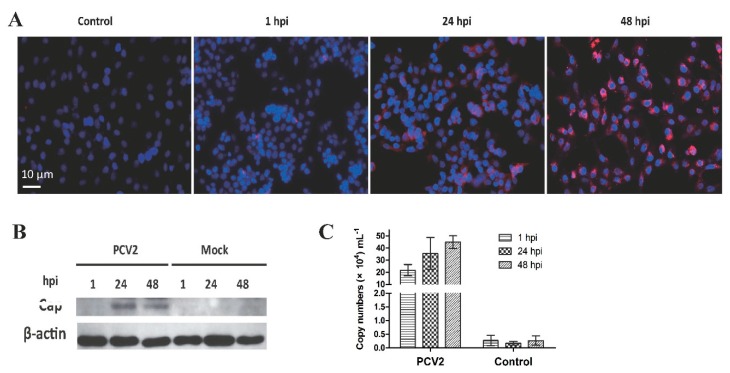
Porcine circovirus type 2 (PCV2) infection in porcine alveolar macrophages (PAMs). (**A**) PAMs were infected with PCV2b and processed for immunofluorescence assay at 1, 24, and 48 hpi. PCV2b was visualized by the ODs-psdAb probe against the Cap protein (red), the nucleus was stained with DAPI (blue). (**B**) Cap protein in PCV2b-infected PAMs was also detected by Western blot assay based on Cap protein (VLPs) immunized porcine serum. (**C**) PCV2 copy numbers were analyzed by qPCR at the time-points indicated. Results are mean ± SE of three independent experiments.

**Figure 2 viruses-11-01021-f002:**
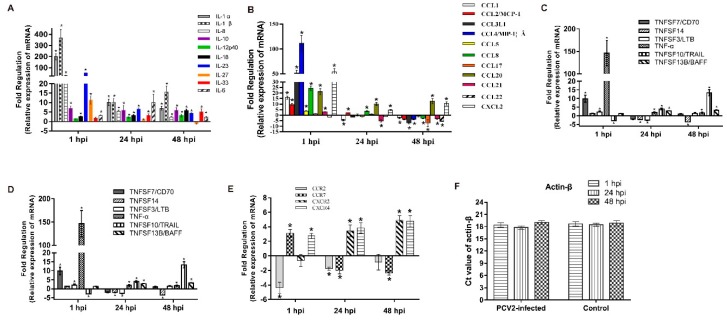
Changes in mRNA expression levels of inflammatory cytokines and receptors. PCR array assay system was used to assess the mRNA expression levels of interleukins (**A**), chemokines (**B**), tumor necrosis factor (**C**), colony-stimulating factors (**D**), and receptors of chemokine (**E**) in PAMs incubated with PCV2 and controls after 1, 24, and 48 h. The Ct value of actin-β in PCV2-infected and control groups (**F**). Fold regulation was obtained by comparing the relative expression intensity of PCV2-infected and uninfected control PAMs. Results are mean ± SE of three independent experiments. * represents *p* < 0.05.

**Figure 3 viruses-11-01021-f003:**
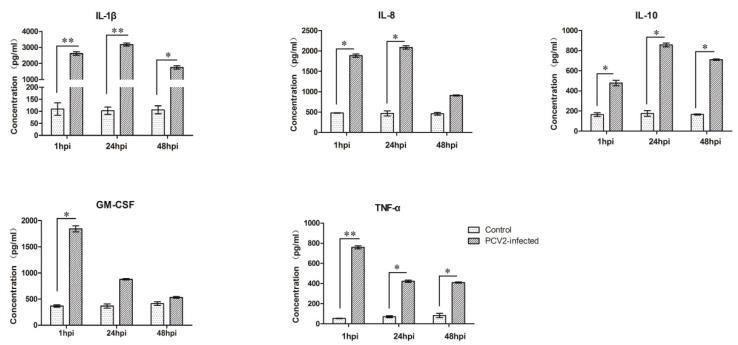
Validation of PCR array data by porcine cytokine array system. PCV2-induced changes in the secreted levels of IL-1β, IL-10, IL-8, GM-CSF, and TNF-α in the culture supernatants of control and PCV2-infected PAMs were measured using a RayBiotech fluorescent Quantibody^®^ porcine cytokine array system. Data are mean ± SE of three independent experiments. * represents *p* < 0.05, ** represents *p* < 0.01.

**Figure 4 viruses-11-01021-f004:**
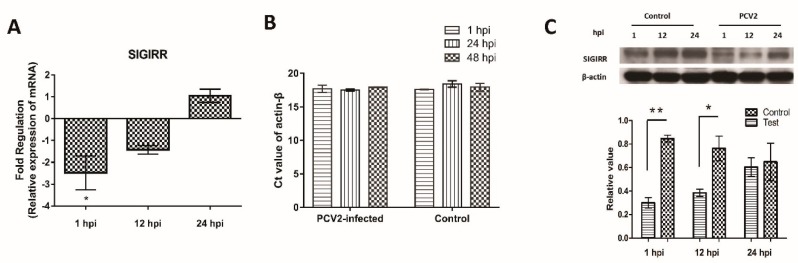
Downregulation of single-immunoglobulin interleukin-1 related receptor (SIGIRR) in PAMs after PCV2 infection. The expression of SIGIRR in PAMs after PCV2 infection were examined using real-time qPCR. The expression of SIGIRR mRNA was normalized to actin-β and presented as fold change relative to the control (**A**). The Ct value of actin-β in PCV2-infected and control groups (**B**). The expression of SIGIRR in protein level was detected by Western blot (**C**). Densitometric analysis was performed to quantify the ratio of SIGGIR to actin-β. Results are mean ± SE of three independent experiments. * represents *p* < 0.05, ** represents *p* < 0.01.

**Figure 5 viruses-11-01021-f005:**
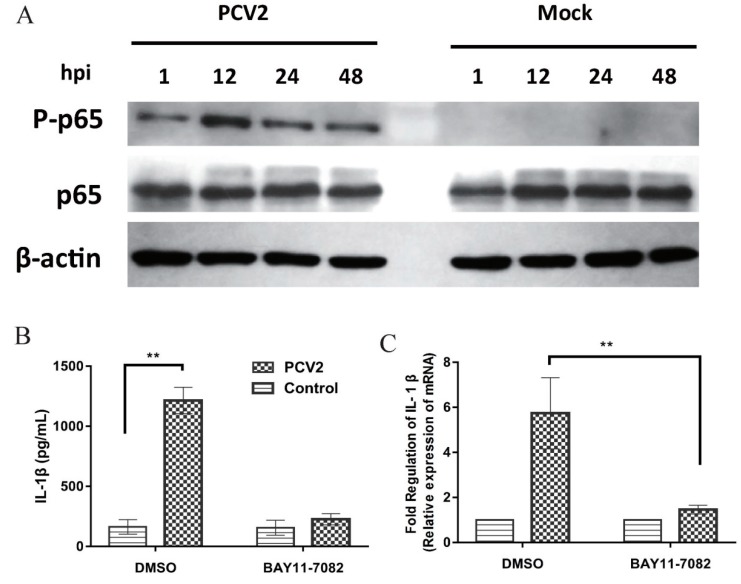
NF-κB activation is critical for PCV2-induced IL-1β production in PAMs. PAMs were infected with PCV2 at an MOI of 1 for 1, 12, 24, and 48 h, and the p-p65 and p65 expression were measured by Western blotting (**A**). PAMs were treated with DMSO, BAY11-7082 (1 µM) for 1 h then infected with PCV2 at an MOI of 1. The supernatants were collected at 12 hpi and analyzed for IL-1β by ELISA (**B**). Total cells were harvested and analyzed for IL-1β mRNA expression by real-time RT-PCR (**C**). DMSO, the solubilization buffer of BAY11-7082, was used as a control. Results are mean ± SE of three independent experiments. ** represent *p* < 0.01.

**Figure 6 viruses-11-01021-f006:**
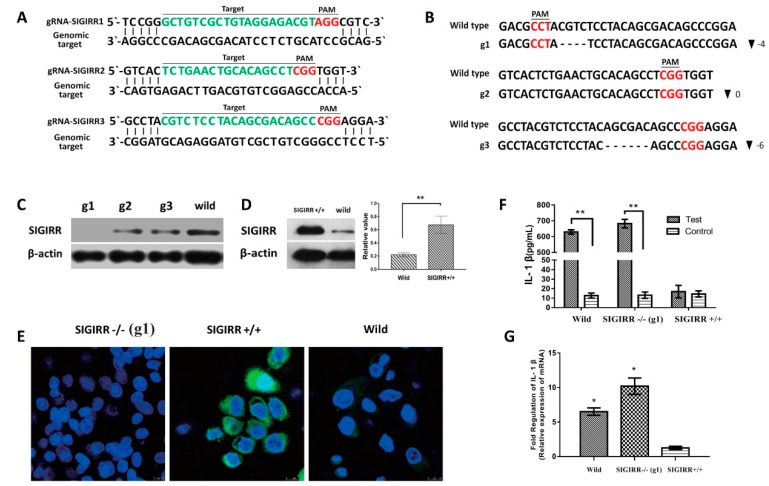
The downregulation of SIGIRR promotes IL-1β overexpression. SIGIRR was knocked off in porcine alveolar macrophage cell line 3D4/2. (**A**) Graphical representation of targeted gRNA at porcine SIGIRR loci. Targeted genomic loci are indicated in green, protospacer adjacent motifs (PAM) sequences are marked in red. (**B**) Sanger sequencing of genomic PCR products encompassing the genomic-editing site from cells g1, g2, and g3. The wild type reference sequence is shown on the top. The bottom is a targeted sequence with deletions. the size of deletions (▼) on the right of each allele. (**C**) Western blotting analysis of SIGIRR protein in cells containing CRISPR-Cas9 system plasmids that were selected by puromycin. (**D**) Western blotting analysis of SIGIRR protein in cells containing expression plasmid pCDH-SIGIRR that was selected by puromycin. (**E**) IFA analysis of SIGIRR in g1 (SIGIRR-/-), overexpression cells (SIGIRR+/+), and wild type cells. (**F**) The IL-1β secretion was measured in wild type, SIGIRR-/-, and SIGIRR+/+ cells at 12 h after PCV2 inoculation. (**G**) The IL-1β mRNA levels were detected in wild type, SIGIRR-/-, and SIGIRR+/+ cells at 12 h after PCV2 inoculation. Results are mean ± SE of three independent experiments. * represent *p* < 0.05, ** represent *p* < 0.01.

**Figure 7 viruses-11-01021-f007:**
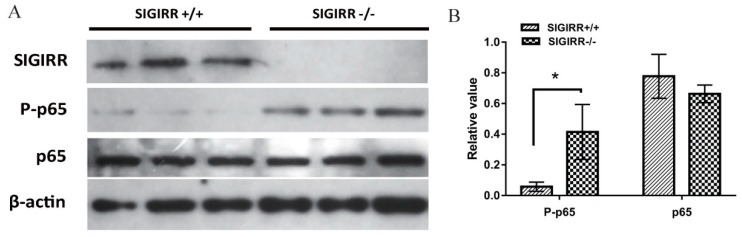
SIGIRR-/- cells have an increased NF-κB activation compare with SIGIRR+/+ cells. SIGIRR-/- and SIGIRR+/+ 3D4/2 cells were infected with PCV2 at MOI of 1 for 12 h, and the SIGIRR, p-p65, and p65 expression were measured by Western blotting (**A**). Densitometric analysis was performed by Image J software and results were normalized to β-actin (**B**). Results are mean ± SE of three independent experiments. * *p* < 0.05.

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
