# Peer review of "Porcine Circovirus Type 2 Induces Single Immunoglobulin Interleukin-1 Related Receptor (SIGIRR) Downregulation to Promote Interleukin-1β Upregulation in Porcine Alveolar Macrophage"

_viruses, 2019, doi:10.3390/v11111021_

Round 1

Reviewer 1 Report

Summary:

Yang et al. study the immune response of porcine alveolar macrophages (PAMs) to Porcine circovirus type 2 (PCV2) infection. On the one side, the authors provide an immune profiling of 84 immune targets; on the other side, they study in detail the mechanisms of PCV2-induced interleukin-1beta (IL-1b) expression. Thereby, the study provides a general overview of the involved immune mechanisms in PCV2-infection and a detailed mechanistic understanding of the important IL-1b response in PCV2-infected PAMs. A strength of the study is that for most analyses, the authors use more than one technique to verify their results. Two limitations of the study are i) the use of only a single housekeeping gene which is substandard; and ii) the lack of confirmation that the PCV2 infection rates of BAY11-7082 and SIGIRR treated PAMs is comparable. The manuscript is well-written and provides a significant understanding of the immune mechanisms following PCV2 infection in PAMs.

Detailed comments:

L. 127: The use of only 1 housekeeping gene is substandard. In addition, Figure 4 B indicates that PCV-2 infection decreases beta-actin expression. If the expression of beta-actin is similar between the MOCK and PCV-2-infected PAMs, please include their Ct values at the different time points as Fig. 2A and make the original Fig. 2A Fig. 2B and so on. If there is a significant difference between the beta-actin levels of infected and MOCK PAMs, the authors should test different housekeeping genes and select a good combination in order to be able to properly compare their immune gene expression levels. Please see J Appl Genet. 2013; 54(4): 391–406. "In the past, validation process was often avoided and HKGs [house keeping genes] were used due to a common belief that they are characterized by constant expression level regardless of the conditions and origins. As awareness of the complex expression regulation networks in the cell function grew, this statement begun to be undermined and experimental confirmation of the stability of candidate genes is now a standard requirement." L. 168: Cloned L. 187: Please state in the text why red fluorescent cells show PCV2, not only in the figure legend. L. 201: Indicate how many genes were analyzed before stating the number of differentially expressed genes. Figure 4B: Beta-actin levels differ substantially between the PCV2-infected and control PAMs. Please describe in the figure legend how the "relative value" was calculated and indicate if the calculation includes a comparison to the beta-actin. Figure 5+6: It is possible that the treatment reduced or inhibited the PCV2 infection in the treated PAMs. This could also explain the difference in the studied immune targets. Therefore, the authors need to show that the infection rates of untreated and the differently treated PAMs was comparable. L. 261/2: Please modify this sentence.

Author Response

Dear reviewer

Thank you very much for your hard work on the manuscript, all the comments help us improve the quality of the manuscript in detail and writing. The authors have been revised the manuscript according to these comments, and all the changes were marked in red letters in the revised manuscript version.

Best Regards,

 Authors

Reviewer 2 Report

This manuscript presents a study on intracellular cytokine upregulation in porcine alveolar macrophages upon infection in vitro with porcine circovirus type 2. The rationale for this study is the importance of porcine circovirus infection for the swine industry, despite the implementation of effective vaccination strategies, and the prime role of lung alveolar macrophages in the infection process. Working with primary cells and the ShDNY-PCV2 PCV2b virus strain, alveolar macrophages from virus-free young pigs showed the induction of a large series of inflammatory cytokines and respective receptors during the first days after infection. This included IL-1beta that was subjected to further studies. The expression and synthesis of this cytokine proved to depend on NF-kappaB activation, as documented by inhibition by BAY11-7082, an inhihibitor of the signal transduction pathway. The expression of single-immunoglobulin interleukin-1 related receptor (SIGIRR), a negative regulator of toll-like receptors mediated immune responses, was downregulated during this first period after infection. This was further investigated in cells from the 3D4/2 alveolar macrophage cell line, by knock-down of SIGIRR using the CRISPR-Cas9 technology, and overexpression of SIGIRR using transfection with a plasmid containing the SIGIRR gene. Studies of these cells after in vitro virus infection revealed that IL1-beta was upregulated in cells lacking SIGIIR, and downregulated in cells overexpressing SIGIIR. The absence of SIGIIR was related with induction of  NF-kappaB activation. It is concluded that the induction of IL-1beta expression in macrophages upon porcine circovirus type 2 infection is related to NF-kappaB activation caused by the downregulation of SIGIRR.

This is an interesting study pointing to role of SIGIIR in activation of signal transduction and subsequent synthesis of inflammatory cytokines, in the process after virus infection of porcine macrophages. The design of experiments is clear, and also the presentation of results. The conclusions are valid.

There are a number of comments to improve the quality of the manuscript.

The studies were conducted on primary alveolar macrophages, and on cells of a macrophage cell line. The differentiation between these two cell types, and the rationale for selection of either one of these cell types is not clear. Also, cytokine arrays and assessment of multiple inflammatory cytokines and receptors was done in primary cells, while more detailed studies on SIGIIT, NF-kappaB activation and IL-1beta expression was done in the macrophage cell line. It appears that the present study essentially is the combination of two studies in one report. This aspect needs clarification. Besides the selection of macrophage cell type, the selection of IL-1beta out of a large series of cytokines needs a rationale. There are no details on the macrophage cell line 3D4/2. These need to be given, including items for which the cell line was not like primary macrophages, and items for which the cell line resembled macrophages. Also it need to be stated to what extent the cell line was negative for viruses. Regarding the manufacturing of primary alveolar macrophages, it need to be stated how many donor animals were used, and what was the quality of the cell preparations. The question is whether there were assays performed to analyze the content and contamination of the ell preparations. The experiments included the use of the inhibitor of the signal transduction pathway, BAY11-7082. There are no data on this compound, only a literature reference. More data on this compound are needed. Section 3.6 needs rephrasing, as it is not completely clear. The first sentence describes the intention to create SIGIIR+/+ and SIGIIR-/- cells: it is advised to separate these two, i.e., create a paragraph on creation of SIGIIR+/+ cells and a separate paragraph on creation of SIGIIR-/- cells. Then the difference between g1 and g3 cells is not clear: both had deletions, but only g1 cells showed he absence of SIGIIR: so the question is raised what happens in the creation of g3 cells. The discussion can be somewhat reduced, with focus on interpretation of data rather than describing perspectives. Regarding the conclusions, the use and limitations of different cell types needs to be mentioned, as well as the broader interpretation of data on IL-1B with respect to the broad range of cytokines and receptors.

Author Response

Dear reviewer

Thank you very much for your hard work on the manuscript, all the comments help us improve the quality of the manuscript in details and writing. The authors have been revised the manuscript according to these comments, and all the changes were marked in red letters in the revised manuscript version.

Best Regards,

 Authors
